# Simultaneous Occurrence of Field Epidemics of Rabbit Hemorrhagic Disease (RHD) in Poland Due to the Co-Presence of *Lagovirus europaeus* GI.1 (RHDV)/GI.1a (RHDVa) and GI.2 (RHDV2) Genotypes

**DOI:** 10.3390/v17101305

**Published:** 2025-09-26

**Authors:** Andrzej Fitzner, Wiesław Niedbalski, Beata Hukowska-Szematowicz

**Affiliations:** 1Department of Virology and Viral Animal Disease, National Veterinary Research Institute—State Research Institute, ST. Wodna 7, 98-220 Zduńska Wola, Poland; wieslaw.niedbalski@piwzp.pl; 2National Reference Laboratory for Rabbit Hemorrhagic Disease (RHD), ST. Wodna 7, 98-220 Zduńska Wola, Poland; 3Institute of Biology, Laboratory of Infectious Biology, Molecular Biology and Immunology, University of Szczecin, St. Z. Felczaka 3c, 71-412 Szczecin, Poland; 4Molecular Biology and Biotechnology Center, University of Szczecin, St. Wąska 13, 71-412 Szczecin, Poland

**Keywords:** Rabbit Hemorrhagic Disease (RHD), *Lagovirus europaeus*, RHDVa, RHDV2, co-occurrence, epidemiology

## Abstract

The highly fatal rabbit hemorrhagic disease (RHD) that first emerged in 1984 in China has spread worldwide and affects both domestic and wild rabbits. The disease was originally caused by RHD virus (*Lagovirus europaeus, L.europaeus*) of GI.1 genotype, but over the years, two further pathogenic forms, known as the antigenic and genetic variant RHDVa (GI.1a) and RHDV2 (genotype GI.2), have been identified. RHD was first reported in Poland in 1988, when two RHDV strains were isolated, currently classified as GI.1c, while RHDVa and RHDV2 emerged in 2003 and 2016, respectively. In this study, using virological and molecular methods, we characterized five new RHDV strains belonging to GI.1 (RHDV)/GI.1a (RHDVa) and GI.2 (RHDV2) genotypes isolated in Poland in 2020–2022, in domestic rabbits from backyard farm and companion animals. We showed that two strains of *L. europaeus* (NRU 2020 and LIB 2020) from 2020 in the phylogenies of nonstructural proteins (NSP) and structural capsid protein (SP-VP60) clustered in a homogeneous GI.1a variant group. We stated that three strains of *L. europaeus* from 2020 to 2022 (KOB 2020, ZWO 2021, WAE 2022) in the VP60 phylogeny were positioned in the GI.2 (RHDV2) genotype, while in the NSP phylogeny, they are genetically related to recombinants with the GI.3/GI.2 genotype. Unexpectedly, in two RHD cases identified in the same small geographical area of south-eastern Poland (Libusza and Kobylanka), the close coexistence of RHDVa (LIB2020) and RHDV2 (KOB2020) strains capable of causing independent infections at the same time was found. This leads to the conclusion that the close natural coexistence of RHDV strains belonging to different genotypes does not necessarily have to directly lead to the emergence of new genetic or antigenic variants, which confirms the distinctness of both genetic forms and indicates different evolutionary paths leading to the best possible adaptation to the host.

## 1. Introduction

Rabbit hemorrhagic disease (RHD), a fatal disease of European wild and domestic rabbits (*Oryctolagus cuniculus*), has been known since 1984, when it was first reported in China [1]. In the late 1980s, RHD spread worldwide, affecting rabbits in Asia, subsequently in Europe, and then in Africa, Central America, North America, as well as Australia and New Zealand [2]. At that time, the disease also spread to Poland and neighboring countries in Central Europe, resulting in high morbidity and mortality among animals, which in turn led to significant economic losses in rabbit breeding [3,4]. RHD is caused by rabbit hemorrhagic disease virus (RHDV), genus *Lagovirus* of the *Caliciviridae* family, which includes the pathogenic European brown hare syndrome virus (EBHSV) and non-pathogenic rabbit and hare viruses [5,6].

*Lagoviruses* are non-enveloped viruses with a polyadenylated genome composed of a single-stranded positive-sense RNA. The genome of RHDV of approximately 7.5 kb consists of two open reading frames (ORFs). ORF1 encodes a large polyprotein consisting of six non-structural proteins and the major capsid structural protein VP60, which forms an icosahedral viral capsid with a diameter of 34–40 nm. ORF2 encodes a smaller structural protein, VP10 [2,7]. Since both RHDV and EBHSV cannot be cultivated in vitro, an alternative to receive their antigens has become virus-like particles (VLPs) produced in various expression systems [8,9,10,11,12,13,14,15]. Recently, it has been demonstrated that chimeric VLPs can simultaneously elicit hemagglutination inhibition antibodies against the GI.1 and GI.2 VP60 proteins, providing effective protection against GI.1 and GI.2 RHDV [16].

Over the years, despite the initially reported antigenic and serotype stability of the virus, confirmed, among others, by the effectiveness of various vaccines containing inactivated RHDV, five genetic groups (G1–G5) were identified among the virus strains based on phylogenetic studies of the VP60 structural protein gene. Some of them gradually disappeared or transformed into new genetic groups [17,18]. Considering the genetic diversity associated with the presence of previously unknown pathogenic and non-pathogenic lagoviruses of rabbits and hares, the new classification based on phylogenetic relationships was presented in 2017 [6], which distinguished four variants within the classic RHDV (GI.1 genotype (RHDV). In 1996, the first non-pathogenic lagovirus of rabbits (RCV), related to pathogenic RHDV, was identified in the gastrointestinal tract of rabbits in Italy [19]. The spectrum of non-pathogenic rabbit lagoviruses is complemented by Michigan rabbit calicivirus (MRCV), detected in the USA, and non-pathogenic rabbit caliciviruses recognized in Europe and Australia, currently known as RCV-E1 (GI.3), RCV-E2, and RCV-A1, which constitute the GI.4 genogroup [6]. In 1996–1997, an antigenic subtype RHDVa, which forms a separate, highly uniform genogroup GI.1a, was identified among pathogenic RHDV strains in Italy and Germany [6,20,21]. The last currently known pathogenic rabbit lagovirus causing RHD is RHDV2 (GI.2 genotype), which was identified in 2010 in France [22,23]. In Poland, *L. europaeus* RHDV2 (RED 2016 and VMS 2017) was first identified in 2018 by Fitzner and Niedbalski [24], and subsequently, the independent introductions of this virus were confirmed (PIN, LIB, and WAK) [25]. This virus, also named RHDVb, affects rabbits of all ages, with a significantly varied mortality range, from relatively low, about 30% to very high, 80–90% [23,26,27,28]. RHDV2, like RHDV and RHDVa, is common throughout the world, but unlike them, it has much broader host specificity and can cause EBHS-like disease in various hare species and infect animals other than lagomorphs [29,30,31,32,33,34,35,36,37,38,39]. In addition, RHDV2, due to far-reaching antigenic and genetic changes, creates a new serotype, as evidenced by the escape from the control associated with the use of a vaccine formulated on classic RHDV. Phylogenetic analyses of RHDV strains since the first cases of RHD in Europe, but also after the release of the RHDV Czech strain in Australia, have indicated its genetic stability [17,40,41]. Meanwhile, early genetic analyses of RHDV have already shown the presence of recombination in the VP60 protein gene of the classical RHD virus strains [42,43], as well as the RHDVa variant [44,45]. However, studies to determine the origin of RHDV2 using whole-length genomic sequences have shown that the GI.2 strains isolated from rabbits, but also those detected in hares, are in fact recombinants whose nonstructural part of the genome derived from the non-pathogenic caliciviruses RCV, RCV-E1 (GI.3), RCV-E2 and RCV-A1 (GI.4), or from the pathogenic lagoviruses, including classical RHDV (GI.1), and RHDVa (GI.1a), as well as from EBHSV (GII.1) [46,47,48,49,50,51,52,53]. To better understand the epidemic, it is crucial to analyze nucleotide sequences covering the complete RHDV2 genome, with particular emphasis on the non-structural regions and the structural capsid proteins, as these elements reveal essential aspects of viral structure and evolutionary pathways.

This paper presents the results of virological, molecular, and phylogenetic analyses of *Lagovirus europaeus* RHDVa and RHDV2 strains identified in rabbits from backyard farms and companion rabbits in southeastern, central, and southwestern Poland from 2020 to 2022, with particular emphasis on the spatial and temporal co-occurrence of infections. Moreover, the examination of the material provided, partly from areas where the occurrence of RHDVa and RHDV2 strains has been confirmed in previous years, allowed a comparative analysis of the persistence of the identified virus and its variability.

## 2. Materials and Methods

### 2.1. Examined Samples

Rabbits’ liver from five cases suspected of RHD reported in 2020–2022 from south-eastern, central, and south-western Poland were analyzed (Table 1, Figure 1).

The biological material originated from rabbits reared in small backyard farms located in rural areas in Libusza (LIB 2020) and Kobylanka (KOB 2020) in Malopolskie Voivodeship, and in the vicinity of the city of Zwoleń (ZWO 2021) in Mazovian Voivodeship, as well as from the pet rabbits from an urban area of Nowa Ruda (NRU 2020) in the Lower Silesian Voivodeship, and from a large urban area in the city of Warsaw (WAE 2022) in Mazovian Voivodeship. The frozen liver was sent by veterinarians from local veterinary clinics to the National Reference Laboratory for RHD, National Veterinary Research Institute—State Research Institute (NVRI), Zdunska Wola, Poland, for diagnostic testing for RHD virus.

Among three cases of RHD reported at the end of July 2020, one was observed in the Lower Silesian Voivodeship (NRU) and two more than 400 km apart in the Malopolskie Voivodeship. In addition, the epizootics LIB and KOB, apart from the temporal convergence, are connected by a very close geographical location of farms (distance less than 5 km), located in two neighboring villages within the same district.

As regards the NRU 2020 outbreak, the biological material came from a five-year-old miniature rabbit living in a small urban environment. Based on medical history, it was determined that three other companion rabbits had died at the exact location shortly before, despite treatment, exhibiting only loss of appetite and nervous signs. The 2020 RHD LIB outbreak involved an entire litter of three-month-old mixed-breed rabbits from a rural farm that had not been vaccinated against RHD. Eight animals died suddenly, presenting with convulsions, shortness of breath, and frothy nasal discharge before dying. The adult female survived the infection. Samples from three rabbits were provided for analysis. The 2020 RHD KOB outbreak originated in a small-scale farm with several litters of four-week-old rabbits. Samples from 8 out of 54 rabbits that died (mortality 93%) were provided for analysis. One female and five young animals survived infection. In a case of RHD ZWO outbreak, a 3.5-month-old rabbit of the New Zealand breed was found dead in January 2021 at a livestock farm where, shortly before, 36 unvaccinated mixed-breed rabbits died within a month (mortality 94.5%). In the case of the 2022 RHD WAE, a small section of rabbit liver suspended in a buffer, taken from a dead 3.5-year-old female companion rabbit from a highly urbanized area of Warsaw, was sent for laboratory diagnosis in October 2022 from a veterinary clinic where an animal was previously treated symptomatically.

### 2.2. Virological Testing (HA, ELISA)

The samples of liver from five cases suspected of RHD were tested using the HA test to evaluate the hemagglutinating characteristics of the virus, as described previously [25]. An ELISA was carried out (ref. code 80415 IZSLER, Brescia, Italy; WOAH Reference Laboratory for RHD) with a set of specific monoclonal antibodies that allow both the detection and differentiation between RHDV, RHDVa, and RHDV2, according to the manufacturer’s protocol.

### 2.3. Molecular Tests

Total RNA was extracted from 100 µL of liver homogenates using a RNeasy Mini Kit (Qiagen, Hilden, Germany) according to the manufacturer’s instructions and used for RHDV genetic material detection by RT-PCR tests. Two real-time RT-PCR methods with specific probes and oligonucleotide primers were used to detect classic RHDV/RHDVa and RHDV2 [54,55]. For detailed molecular characterization of the analyzed strains, genome fragments covering nonstructural genes, VP60 and VP10 structural genes, and 3′UTR were amplified using the OneStep RT kit (Qiagen, Hilden, Germany) and specific primer pairs of oligonucleotide primers, as described previously [24]. PCR amplicons were visualized in a 1.5% agarose gel, purified, and directly sequenced in both directions using the ABI Prism BigDye Terminator v3.1 Cycle Sequencing Kit on an ABI 3730XL DNA sequencer (Life Technologies, Carlsbad, CA, USA) at the Genomed Joint-Stock Company sequencing service (Warsaw, Poland). Five RHDV genome sequences encoding ORF1, ORF2, and 3′UTR received in this study were submitted using the BankIt 3.0 tool to GenBank and registered under the following accession numbers: OQ605827, OQ605828, OQ605829, OQ605830, and OR488784.

For comparative analysis and evaluation of nucleotide identity of the received sequences, BLAST software was used (accessed on 1 August 2025, https://blast.ncbi.nlm.nih.gov/Blast.cgi) [56,57]. The VP60 gene and the nonstructural (NS) part of the genome sequences of five RHDV strains were analyzed, and RHDV sequences from Europe, Asia, Australia, and North America available in databases were used for phylogeny. The RHDV sequences obtained in this study were aligned with the 111 VP60 gene sequences and 108 NS sequences, being representative of different genogroups of pathogenic RHDVs (GI.Ib, c,d; GI.1a; GI.2) and non-pathogenic rabbit lagoviruses (GI.3, GI.4, MRCV, RCV) using ClustalW [58]. We also included sequences of Polish RHDV, RHDVa, and RHDV2 strains already detected in farmed and pet rabbits between 1989 and 2018. The GenBank accession numbers are indicated in the Appendix A and Figure 2 and Figure 3. The sequence of the EBHSV reference strain (EBHSV-GD, accession number Z69620) was used as an outgroup to root the trees. Phylogenetic trees were constructed in MEGA7 [59] using a Maximum Likelihood (ML) method with the Tamura–Nei model with uniform rates, and the Nearest-Neighbor-Interchange (NNI) method for topology searching. A branch support was estimated using 1000 bootstrap replicates. Bootstrap values ≥ 70% were considered indicative of moderate support.

## 3. Results

### 3.1. Detection of Lagovirus europaeus (RHDVs) by HA, ELISA, and Real-Time RT-qPCR

The results of the HA test showed a strong hemagglutinating activity of NRU 2020, LIB 2020, KOB 2020, and ZWO 2021 isolates with titers ranging from 1:10.240 to 1:20.480. In the case of the WAE isolate from 2022 obtained from a pet rabbit, no hemagglutination was observed.

RHDVs antigen was detected using differential ELISA with MAbs-CR in rabbit liver homogenates from the five outbreaks suspected of RHD. The RHDV isolates NRU 2020 and LIB 2020 were positive with MAb RHDVa, while the isolates KOB 2020 and ZWO 2021 were positive with MAb RHDV2. In case of WAE 2022, no isolate reaction was observed in the ELISA test with any of the MAbs dedicated to RHDVs typing.

Using RT-qPCR with RHDV/RHDVa-specific probe and primers, the RNA presence of classical RHDV/or RHDVa was detected in field samples from NRU and LIB cases of 2020. The CT values in the range of 15–17 indicate a high concentration of the virus’s genetic material, characteristic of acute infection. Three samples showed positive by RHDV2 RT-qPCR, with CT values ranging from 14.2 to 18.5 for KOB 2020, ZWO 2021, and WAE 2022 isolates, respectively. No reactivity was shown for the NRU and LIB isolates of the 2020 outbreaks. The specificity of the results obtained by both RT-qPCR methods was confirmed in reference RNA samples of Polish RHDVa strains and Polish and French RHDV2 strains (CT values 14–15).

### 3.2. Sequence Analysis of L. europaeus (RHDVs) Strains 2020–2022

The amplicons’ sequences were merged into the nearly complete sequences where five Polish *Lagovirus europaeus-GI (RHDVs)* genomes tested had a length of 7428 nt (2020 NRU, 2020 LIB), 7438 nt (2020 KOB, 2022 WAE), and 7439 nt (2021 ZWO) without the initial fragment upstream of the 5ʹ untranslated region. According to BLAST comparison, three Polish RHDV2 field strains (KOB 2020, ZWO 2021, WAE 2022) showed 84–85% nucleotide sequence homology with the genome of the RHDV reference strain (M67473), approximately 84% with RHDVa Triptis (EF558583), and 92–96% with RHDV2 N11 (KM878681). Among these, the highest nucleotide sequence identity of 98,5% was presented by the RHDV 2022 WAE and 2020 KOB isolates. The same genetic distance was observed throughout the full genome, in its non-structural part, and in the VP60 coding gene of these strains. The similarity of RHDV2 2021 ZWO with the 2022 WAE and 2020 KOB sequences in the same parts of the genome is in the range of 90.5–94%. Two RHDVa isolates (NRU and LIB) from July 2020, responsible for separate RHD outbreaks in the South-West and South-East regions of the country, showed 96–97%, 88–92%, and 82–85% nucleotide sequence identity with the complete genome, NS, and VP60 parts of the RHDVa Triptis, RHDV FRG 1989, and RHDV2 N11 strains, respectively. The sequence similarity of two RHDVa strains analyzed in 2020 to the three RHDV2 isolates from 2020, 2021, and 2022 is 84% in the entire genome and its non-structural part and 82% in the VP60 gene, which is consistent with the overall homology observed between RHDVa and RHDV2. RHDVa NRU 2020 and RHDVa LIB 2020 strains share 97.5% nucleotide sequence identity, regardless of the part of the genome analyzed, and show overall high homology with previous Polish and foreign GI.1a strains, with the differences in genetic distance of the VP60 gene ranging from 0.5 to 5.0%. The overall genetic variability based on the evolutionary divergence (distance matrix) between 111 sequences encoding VP60 capsid protein gene and 108 sequences encoding non-structural proteins of the genome is presented in Appendix A.

### 3.3. Phylogenetic Tree Analyses

Phylogenetic analysis of VP60 sequences (Figure 2) showed that two isolates, NRU 2020 and LIB 2020, clustered with GI.1a (RHDVa) strains, while three other isolates (KOB 2020, ZWO 2021, WAE 2022) clustered with GI.2 (RHDV2) genotype.

Furthermore, both analyzed RHDVa isolates, despite a low nucleotide genetic distance (2.2%), were separated into two subgroups, where the LIB 2020 isolate clustered with older Polish RHDVa strains detected in the same geographical area seven years earlier, GLE 2013 (KY319032) and STR2 2013 (KY679904), while the NRU 2020 isolate clustered with the Chinese WHNRH 2005 (DQ280493) and the USA In-05 (EU 003578) strains of 2005. The close relationship between LIB 2020 and GLE 2013 strains within the same sub-group is confirmed by the 97% bootstrap value.

Regarding the isolates identified as RHDV2, VP60 phylogeny revealed that KOB 2020 and WAE 2022 clustered with the subgroup represented by the French isolate 16–36 OOd 2016 (MN738377), as confirmed by a very high bootstrap value of 100%, which also includes the Polish RHDV2 strain LIB 2018 (MN853659). In turn, strain ZWO 2021 was positioned separately at the end of the overall RHDV2 clade. Phylogenetic analysis of the nonstructural genes (Figure 3) also showed a clear division of the five analyzed strains into distinct genetic clusters, according to which the NRU 2020 and LIB 2020 strains grouped again in the RHDVa clade (GI.1a), while the three remaining strains clustered with lagoviruses currently identified as GI.3-GI.2 recombinants. Similarly to the VP60 phylogeny results, the KOB 2020 and WAE 2022 strains showed a close relationship to the French RHDV2 16–35 OOd 2016 isolate (MN738377). They were positioned in a broader subgroup with RHDV2 strains identified in 2019 in Mexico (OM973948) and in 2020 in the United States (MT506237) (Figure 3). In turn, the RHDV2 ZWO 2021 strain was assigned to the GI.3-GI.2 recombinants, the closest of which is the 2019 Tunisian strain (MZ913394) and the 2011–2012 RHDV2 strains from the Iberian Peninsula.

### 3.4. Overview of Five New Lagovirus europaeus–GI.1 (RHDV) and GI.2 (RHDV2) Genotypes Identified in Domestic and Companion Rabbits in Poland in 2020–2022 (Table 2)

The virological and molecular characteristics and phylogenetic relationships of *Lagovirus europaeus* strains isolated in Poland in 2020–2022 are presented in Table 2.

## 4. Discussion

Rabbit hemorrhagic disease and European brown hare syndrome, two highly contagious viral diseases of hares and rabbits, which appeared in the early 1980s in Europe and Asia, immediately became a massive threat to the populations of free-living and farmed hares (*Lepus europaeus* and *Lepus timidus*) and farmed and wild domestic rabbits (*Oryctolagus cuniculus*) [2,61]. RHDV strains isolated from farm and wild rabbits in the first period after the disease was recognized in Europe were characterized by antigenic and genetic stability, being serotypically homogeneous [18,40]. The process of RHD virus evolution was confirmed by the detection in 1996 in Italy, and slightly later in Germany, of the antigenic and genetic variant RHDVa (GI.1a), the strains of which were detected on several continents and form a uniform genetic group G6 [20,21]. Clearly distinct genetic and antigenic features are presented by the pathogenic lagovirus RHDV2 (GI.2), first detected in 2010 in France and in 2011 on the Iberian Peninsula [22,23,26].

In general, the epidemiology of RHD in Poland reflects the key moments related to the emergence of this deadly disease in Europe and the activity of the three currently known pathogenic forms of the RHDV. RHD was first reported in Poland in 1988, when the oldest indigenous RHDV strains SGM and KGM of genotype GI.1c were isolated [3]. At the same time, very similar genetically RHDV strains with hemagglutinating properties were commonly isolated from rabbits that died during mass epidemics reported in other Central European countries [4,62,63,64,65]. It should be noted that the presence of classic RHDV GI.1b strains has never been detected in Poland, whereas GI.1d isolates started to appear around 1994 and were detected for at least the following 10 years [66]. However, to better understand the epidemiology of RHD in Poland, it should be stated that, unlike the classic RHDV, the RHDVa variant strains appeared in Poland several years later than in Italy [20] or in neighboring Germany [21], from where they most probably affected our country between 2003 and 2004 [67,68]. The occurrence of the RHDVa variant in this part of the continent was confirmed at a similar time in Hungary [64] and Russia [65]. The GI.2 genotype appeared in our country with a delay of several years compared to France and other countries from Western Europe [2,24]. The genetic profile of the first two Polish RHDV2 strains isolated in 2016 and 2017, and the phylogenetic closeness with the group of RHDV2 GI.1b-GI.2 recombinants from the Iberian Peninsula from 2013 to 2015, including the German sequence (LR899189) from 2014, clearly show the origin of the virus. Since there is no information on RHDV2 strains from other Central European countries, the Polish RHDV2 sequences remain the only reference point for assessing the viral distribution routes and variability of the virus strains in the region, as well as in the continental and global dimensions.

To complete the picture of the epidemiology of RHD in Poland and the related background, it is also important to note that the main environment in which the virus can easily spread and persist is the extraordinarily numerous, small backyard farms, where open breeding, exchange of breeding material between breeders, and the use of fresh feed in the summer season are standard practices. Given the relatively low number and share of large industrial farms in the country’s rabbit breeding sector, the risk of introducing the disease is also low, thanks to high biosecurity standards. In turn, wild rabbits also seem to be of relatively marginal importance in the spread of this disease in Poland, because their small population is limited to individual enclaves in the central part of the country. However, despite these limitations, our recent studies have shown that the 2015 RHDVa field strain detected in native wild rabbits is among the most diverse Polish GI.1a strains [25]. There are many indications that companion rabbits may play an increasingly important role in the epidemiology of RHD, as evidenced by the recent cases of RHD detected in 2019 in Wrocław, in which RHDVa (GI.1a) and classical RHDV strains of GI.1a genotype were identified [69]. While the detection of RHDVa (GI.1a) in 2019 in southwest Poland (RHDVa NRU strain from this study was confirmed in the same voivodeship a year later) was predictable, the re-emergence of classic RHDV is surprising, as GI.1c group strains have not been detected in Poland since the mid-1990s. However, since only a short fragment of the VP60 sequence of these isolates was determined [69], it cannot be ruled out that the detected viruses are recombinants, and more detailed studies are needed to explain their origin.

In this study, we characterized five new RHDV strains isolated in Poland in 2020–2022. We confirmed the isolation of two GI.1a (RHDVa) and three GI.2 (RHDV2) strains in domestic rabbits from breeding farms and companion rabbits.

**RHDVa.** The two RHDVa strains, NRU and LIB, detected in late July 2020 in the southwestern and southeastern regions of the country, approximately 400 km apart, are genetically very similar, although not identical. Both strains revealed genetic variability levels of 2.2% and 2.6% in VP60 and nonstructural proteins, respectively. In the ML phylogenetic trees of NS and VP60, the 2020 NRU and 2020 LIB sequences were shown to cluster in the homogeneous GI.1a group, positioning themselves in a smaller subgroup (bootstrap value 100% and 86%) represented by the one of the oldest RHDVa strains that was detected in 1996 in Germany (EF558581), the American isolate In-05 (EU003578), the Chinese isolate WHNRH 2005 (DQ280493), and six Polish RHDVa strains detected in the years 2004–2017. Moreover, in both phylogenies, the strain closest to the RHDVa LIB 2020 isolate is the Polish RHDVa strain GLE (KY319032) (genetic distance 0.5–0.9%), which was detected in the same geographical area seven years earlier. The low genetic diversity of RHDVa strains circulating in Poland, compared to RHDV2, is not surprising in light of the general homogeneity of GI.1a variant isolates detected so far worldwide. In RHDVa isolates, intergenomic recombinations of the GI.4–GI.1a type have been detected so far in Australia [44]. In turn, recombinants between G2 (GI.1c) and G6 (GI.1a) within the VP60 gene have been confirmed [45], and between GI.1a–GI.2 in the RdRp-VP60 region on rabbit farms in China [70].

**RHDV2.** Conversely, in the VP60 phylogenetic tree, three RHDV 2020–2022 isolates identified as RHDV2 by ELISA with monoclonal antibody typing also clustered into a large single GI.2 group. This group includes Polish RHDV2 strains from 2016 to 2018 and other RHDV2 sequences found in Europe, Asia, North America, Africa, and Australia. The phylogeny of the 5′ region of the genome coding for non-structural proteins reveals a much more diverse picture of genetic relationships. According to this, all new Polish RHDV2 strains fall into a cluster that contains non-pathogenic lagoviruses GI.3 and recombinant strains GI.3–GI.2, which, based on recent studies, also include the oldest RHDV2 isolates (RHDVb), previously thought to be non-recombinants [50]. The presence of the NS segment—related to the non-pathogenic calicivirus RCV-E1 of European origin—in the RHDV2 genome is well documented in sequences of RHDV2 isolates diagnosed recently in Africa, the United States, Canada, Mexico, Japan, New Zealand, and many European countries. This demonstrates, on one hand, the process of virus differentiation, and on the other hand, the formation of recombinant forms that ensure its survival [50]. Furthermore, since classical GI.1b strains have never been diagnosed in our country, this analysis confirms the external origin of the first native RHDV2 strains from 2016 to 2017 and the emergence of recombinant RCV-E1-RHDV2 strains since 2018.

Among the three new indigenous RHDV2 strains, the ZWO 2021 sequence that branches the GI.2 cluster in the VP60 ML tree appears to be phylogenetically the most distant from the two new RHDV2 strains and other Polish GI.3–GI.2 sequences detected since 2018 (a genetic variability of about 5–6% in VP60 and about 9% in the NSP region) suggesting a distinct viral incursion and origin of this strain. In ML phylogeny of non-structural proteins, this sequence is strongly related with the Tunisian RHDV2 isolate Touza 1 from 2019 (MZ913394) with 4.1% of genetic variability, the Canadian strain WN-AH-2016 (KY235675) (4.8% of genetic variability) and a group of the oldest strains from the Iberian Peninsula: N11, Zar-11-11, Seg08-12 and 16PLMI 1 (genetic variability about 3.5%). In turn, the KOB 2020 and WAE 2022 sequences are related to the French RHDV2 10–28 strain from 2010 and a group of RHDV2 strains from the United States, Mexico, and Canada detected in 2019–2020. However, the most significant similarity connects them with the Polish RHDV2 LIB strain from 2018. These results provide evidence of the spread of this RHDV2 lineage over a large area, covering countries and continents, regardless of the type of rabbit utility, but also suggest the stability of this virus over the years. It is worth emphasizing that the RHDV2 2020 KOB and 2018 LIB strains originate from the same geographical region, indicating the persistence of a genetically stable virus. This is supported by the low levels of genetic variability observed in VP60 (0.4%) and NS (0.6%), as detailed in Appendix A.

Notably, both the VP60 and NS phylogeny clearly showed that the RHDV2 2022 WAE strain has no direct relationship to the four-years-younger RHDV2 WAK strain (MN853661), indicating independent introduction pathways for both isolates, even though the same urban veterinary clinic submitted both. The results of our phylogenetic studies using RHDV complete genome sequences, including sequences representative of different recombinant forms of RHDV2, indirectly indicate that strains 2020 KOB, 2021 ZWO, and 2022 WAE are also GI.3-GI.2 recombinants.

In addition to the diagnostic advantages resulting from the isolation of new viral strains and the determination of their antigenic and genetic characteristics, some of the field cases analyzed here have proven to be particularly interesting from an epidemiological point of view due to the spatiotemporal interdependence of epidemics caused by pathogenic RHD lagoviruses of different genotypes. For the first time, in field conditions, we have demonstrated the close coexistence of RHDVa and RHDV2 strains, which caused independent infections, coincidental in time. Quite unexpectedly, two RHD strains of 2020 of the genotype GI.1a and the genotype GI.2 have been isolated from two cases, identified at the same time in neighboring villages approximately 3 km apart in southeast Poland. As discussed above, RHDVa 2020 LIB and RHDV2 2020 KOB isolates show a very high level of homology, regardless of the analyzed part of the genome, to viral strains of the same antigenic profile, which were previously diagnosed both in the same geographical area and in other regions of the country. This observation indicates the lack of direct interaction of both pathogenic lagovirus types at the genome level, which could result in the formation of modified viral forms.

Given our findings on the generally close phylogenetic relationships between older and younger indigenous strains of RHDVa and RHDV2, we can conclude that these viruses are well adapted to persist in a stable form under specific environmental conditions. This viral fitness allowed indigenous strains of RHDVa and RHDV2 (in the form of GI.3-GI.2 recombinants) to persist for a long time quite independently in the rabbit population raised in traditional backyard farms, as well as to spread among pet rabbits. With the increasing presence of GI.1a variant strains in Poland since 2003, the displacement of classical RHDV (GI.1d) was observed. In turn, the appearance of RHDV2 strains in 2016 resulted in RHDVa and RHDV2 being the main pathogenic lagoviruses detected continuously in new cases of the disease. For many years, they have posed a direct threat to farm rabbits and rabbits kept as companion animals.

Given the threats associated with the persistence of the RHD epidemic worldwide, the study of lagovirus variability remains an essential tool for effective disease prevention and improved diagnostics. As shown by numerous studies, it is the non-pathogenic lagoviruses of rabbits and hares that play a significant role in shaping the genetic diversity of pathogenic caliciviruses via recombination [50,51]. The patterns identified so far indicate recombination between pathogenic RHDV2 and non-pathogenic RCV (GI.4eP-GI.2, GI.4cP-GI.2, GI.3P-GI.2), leading to the formation of even triple recombinants [52]. As a result of genome reconfiguration, recombinants retain pathogenicity associated with structural genes, while nonstructural genes derived from non-pathogenic lagoviruses are supposed to be responsible for the epidemiological fitness [71]. This diversity is particularly well documented in the case of RHDV2 strains. Therefore, analyzing nucleotide sequences spanning the full genome of isolated viral strains aids understanding of their structure and tracking of their evolution, which in turn allows for a better understanding of the nature of the epidemic. However, it should be stated here that there are still many unfilled gaps in the molecular epidemiology of RHD. Our study provided five additional sequences of the RHD virus genome, which include its non-structural part and structural capsid proteins, partially meeting these expectations. It should also be noted that, in the absence of new complete RHD virus sequences from other Central and Eastern European countries, particularly those concerning RHDV2, our results are also a source of information on the epidemiology of the disease, strain variability, as well as the routes of RHD virus spread in this part of the continent.

## 5. Conclusions

The results of our study confirm the persistence of two pathogenic *L. europaeus* genotypes (GI.1/GI.1a and GI.2) as the leading causes of RHD infections in rabbits in Poland, thus confirming their genetic diversity and the recombinant nature of the currently circulating RHDV2 strains. Based on the analysis of RHD field cases of 2020 due to the GI.1a and GI.2 viruses, we conclude that the close environmental coexistence of the genotypes GI.1a and GI.2 does not lead directly to the emergence of new genetic or antigenic variants, suggesting different evolutionary pathways leading to the best possible adaptation to the host. The obtained results provide essential information on the complex epidemiology and evolution of *Lagovirus europaeus* in Poland and emphasize the need for further monitoring of lagoviruses circulating in breeding and companion rabbit populations.

## Figures and Tables

**Figure 1 viruses-17-01305-f001:**
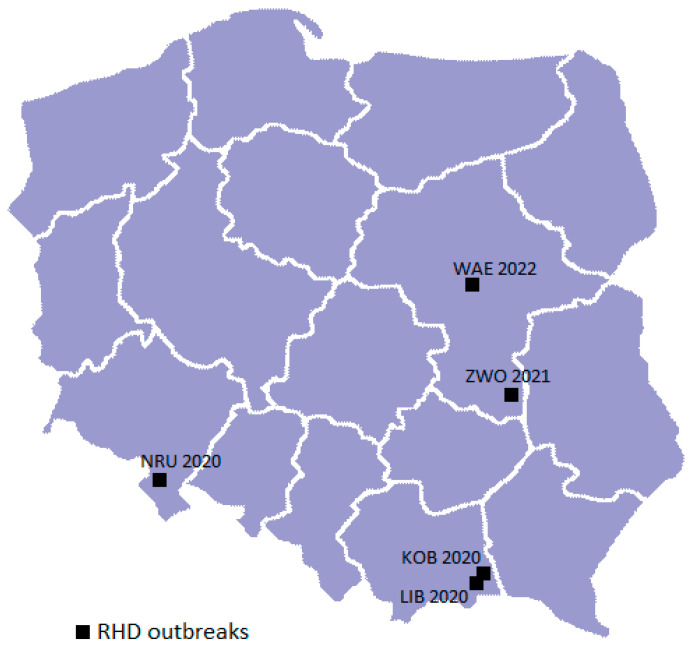
RHD outbreaks location.

**Figure 2 viruses-17-01305-f002:**
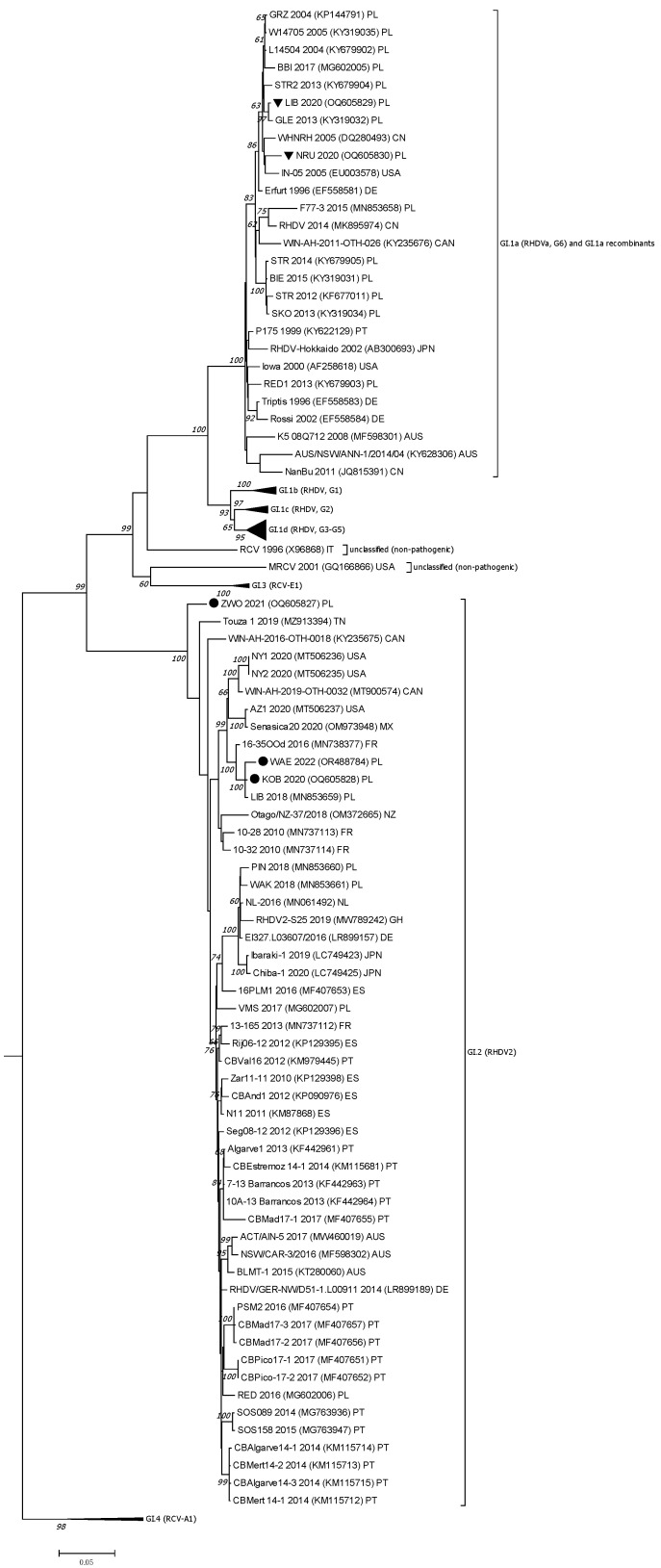
The phylogenetic ML tree for the structural capsid protein gene-VP60 (nucleotides 5305–7044) of lagoviruses sequences (*n* = 111) based on the Tamura–Nei model [60]; bootstrap values (1000 replicates) greater than 60% are shown at the corresponding tree nodes. The tree is drawn to scale with branch lengths in the same units as those of the evolutionary distances used to infer the phylogenetic tree. Evolutionary analyses were conducted in MEGA7 [59]. The European brown hare syndrome virus (GII.1), EBHSV-GD sequence (Z69620) was used as an outgroup to root the tree. The RHDVa sequences received in the study are marked by a black triangle, while RHDV2 sequences are indicated with a black dot. GenBank accession numbers of the sequences are displayed next to their names and are listed in the Appendix A. GenBank accession numbers of the sequences occurring in the collapsed branches, representing genetic variants (do not comprise the sequences of studied RHDV strains), are given in the Appendix A.

**Figure 3 viruses-17-01305-f003:**
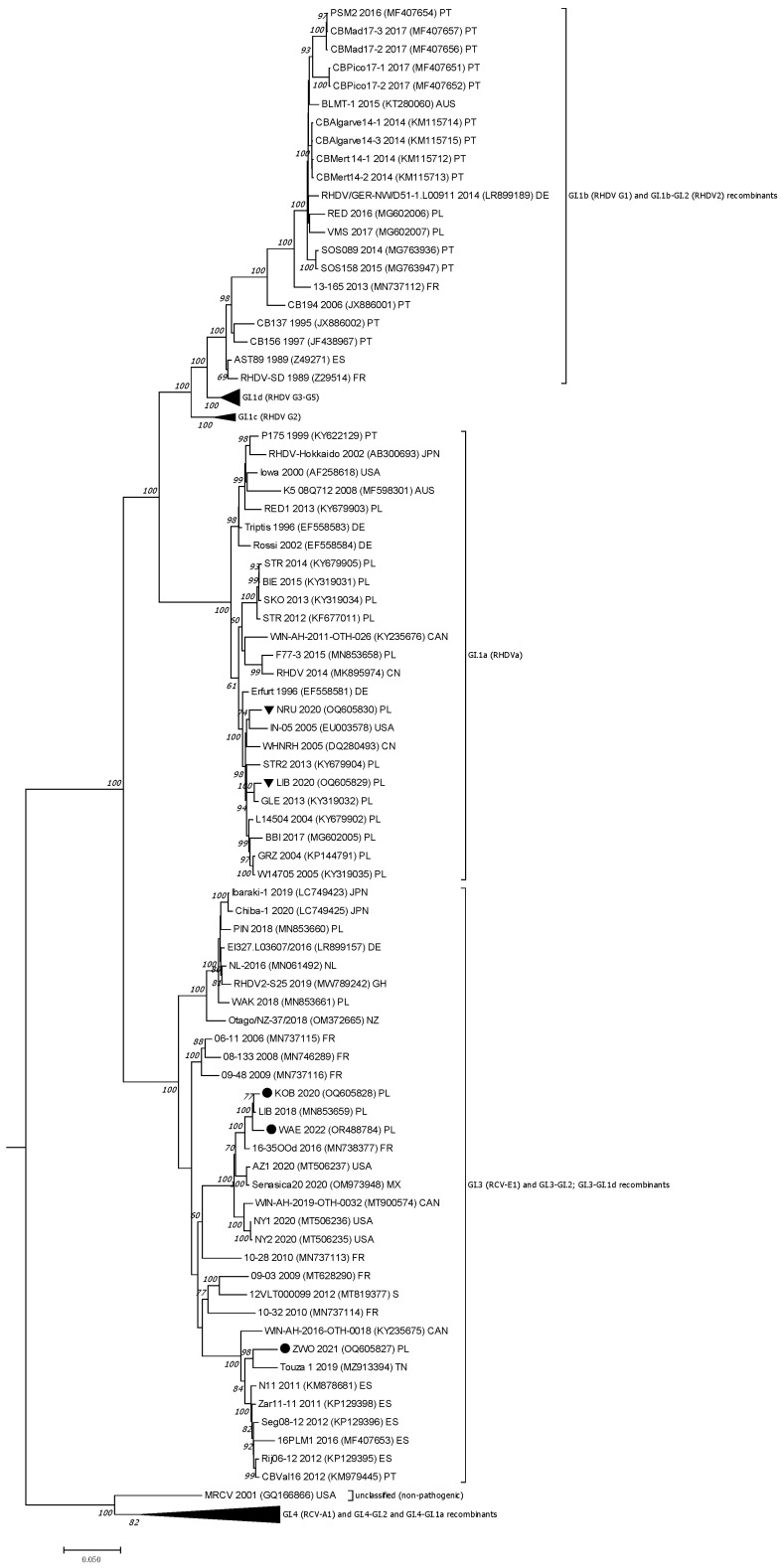
The phylogenetic ML tree for the nonstructural protein genes–NS (nucleotides 20-5304) of lagoviruses sequences (*n* = 108) based on the Tamura–Nei model [60]; bootstrap values (1000 replicates) greater than 60% are shown at the corresponding tree nodes. The tree is drawn to scale with branch lengths in the same units as those of the evolutionary distances used to infer the phylogenetic tree. Evolutionary analyses were conducted in MEGA7 [59]. The European brown hare syndrome virus (GII.1), EBHSV-GD sequence (Z69620) was used as an outgroup to root the tree. RHDVa sequences received in the study are marked by a black triangle, while RHDV2 sequences are indicated with a black dot. GenBank accession numbers of the sequences are displayed next to their names and are listed in the Appendix A. GenBank accession numbers of the sequences occurring in the collapsed branches, representing genetic variants (do not comprise the sequences of studied RHDV strains), are given in the Appendix A.

**Table 1 viruses-17-01305-t001:** Characteristics of five rabbit hemorrhagic disease (RHD) outbreaks caused by *Lagovirus europaeus-GI (RHDVs) genogrup* in Poland between 2020 and 2022.

RHD Outbreaks	Location/Voivodeship	Environmental Type	Data on Dead Rabbits	Vaccination Status
LIB 2020	Libusza, Malopolskie	Rural, farm	3-month-old mixed-breed rabbits	No
KOB 2020	Kobylanka, Malopolskie	Rural, farm	4-week-old rabbits	No information
NRU 2020	Nowa Ruda, Dolnośląskie	Small town	5-year-old miniature companion rabbit	No information
ZWO 2021	Zwoleń, Mazovian	Rural, farm	3.5-month-old rabbit of New Zealand breed	No
WAE 2022	Warsaw, Mazovian	Urban area	3.5-year-old companion rabbit	No information

**Table 2 viruses-17-01305-t002:** Virological, molecular, and phylogenetic characteristics of the new *Lagovirus europaeus*–GI.1 (RHDV)/GI.1a (RHDVa) and GI.2 (RHDV2) genotypes identified in Poland.

Characteristics		*Lagovirus europaeus* Strain				
		NRU 2020 OQ605830 *	LIB 2020 OQ605829 *	KOB 2020 OQ605828 *	ZWO 2021 OQ605827 *	WAE 2022 OR488784 *
HA		(+)	(+)	(+)	(+)	(-)
ELISA (MAbs)		RHDVa (+)	RHDVa (+)	RHDV2 (+)	RHDV2 (+)	(-)
RT-qPCR		RHDV/RHDVa (+) RHDV2 (-)	RHDV/RHDVa (+) RHDV2 (-)	RHDV2 (+) RHDV/RHDVa (-)	RHDV2 (+) RHDV/RHDVa (-)	RHDV2 (+) RHDV/RHDVa (-)
Phylogenetic group/Genotype	SP-VP60	Cluster with RHDVa strains (WHNRH 2025 (Chinese), IN-05 2005 (USA))	Cluster with old Polish RHDVa strains (GLE 2013, STR 2013)	Cluster with RHDV2 strains (16-35OOd 2016, France), and WAE 2022, LIB 2018 (Polish))	Clade RHDV2 strains	Cluster with RHDV2 strains (16-35OOd 2016 (French), and KOB 2020, LIB 2018 (Polish))
		GI.1 (RHDV)/GI.1a (RHDVa)	GI.1 (RHDV)/GI.1a (RHDVa)	GI.2 (RHDV2)	GI.2 (RHDV2)	GI.2 (RHDV2)
	NSP	Clade RHDVa strains	Clade RHDVa strains	Cluster RHDV2 strains (16-35OOd 2016 (France), 2019 Mexico, AZ1 2020 (USA))	Cluster with Tunisian strain Touza 1 2019 (GI.3–GI.2 genotype), and RHDV2 strains from 2011 to 2012 in the Iberian Peninsula	Cluster RHDV2 strains (16-35OOd 2016 (France), 2019 Mexico, AZ1 2020 (USA))
		GI.1 (RHDV)/GI.1a (RHDVa)	GI.1 (RHDV)/GI.1a (RHDVa)	Features of recombinants GI.3/GI.2	Features of recombinants GI.3/GI.2	Features of recombinants GI.3/GI.2

SP-VP60—structural protein-VP60; NSP—non-structural protein; G—genotype; * GenBank accession number.

## Data Availability

The original contributions presented in the study are included in the article; further inquiries can be directed to the corresponding author: Andrzej Fitzner; andrzej.fitzner@piwzp.pl; Beata Hukowska-Szematowicz; beata.hukowska-szematowicz@usz.edu.pl.

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
