# Peer review of "Simultaneous Occurrence of Field Epidemics of Rabbit Hemorrhagic Disease (RHD) in Poland Due to the Co-Presence of Lagovirus europaeus GI.1 (RHDV)/GI.1a (RHDVa) and GI.2 (RHDV2) Genotypes"

_viruses, 2025, doi:10.3390/v17101305_

Round 1
Reviewer 1 Report
Comments and Suggestions for Authors
The paper describes epidemiological studies on Lagovirus europaeus in the country of Poland. This study is important to provide a more complete picture of the epidemiology of this virus. The work was well-preformed and is technically sound. I have some minor, but important, comments and suggestions for improvement of the presentation of this work.
- Abstract is far too long. Some introductory details should be deleted. The results are presented in a too detailed way (as for Abstract). Please, include only crucial results and present major conclusions here.
 - The term 'non-pathogenic virus(es)' is used throughout the paper. Although this term was used also in previous papers, I found it very unfortunate. If fact, from a point of view of a biologist, there are no non-pathogenic viruses. Each virus can propagate only in the cells of its host, thus, each virus must be pathogenic, othervise it could not exist. Obviously, symptoms of infections by various viruses exprerssed by host organisms can vary significantly, from severe to mild, or even asymptomatic infections can occur. However, in no case a virus can be considered non-pathogenic. In my opinion, if the authors want to keep such a nomenclature in this paper, they should either provide such a comment in the text (if they agree with the arguments presented in this review) or provide arguments (in response to review) why they do not agree with my opinion.
 - Figure 2 is too small; especially letters are so small that it is impossible to read any name. This must be corrected, otherwise the figure is useless.
 - Table 2 is constructed against the common rules. The first column should be titled "Characteristic", while "Lagovirus europaeus strain" should be a common heading for columns 2-5.
 - Chapter Conclusions should contain only real conclusions, not a summary of the obtained results. Please, restrict this chapter to a few points indicating what did you conclude on the basis of the obtained results, not providing another description of the whole work (such a description is already provided in Materials and Methods, Results, and Discussion sections, while a summary should be presented in Abstract, rather than in Conclusions).

Author Response
Response to Reviewer 1 Comments
We sincerely thank the Reviewer for helpful comments, which significantly improved the manuscript. We have introduced changes following the Reviewer suggestions. We hope that with these changes, the manuscript will be acceptable for publication in “Viruses”.
Below, we provide a step-by-step response to the comments in the review. Changes in the manuscript are marked in yellow. We appreciate the Reviewer's time and effort on the manuscript.
Point 1: Abstract is far too long. Some introductory details should be deleted. The results are presented in a too detailed way (as for Abstract). Please, include only crucial results and present major conclusions here.
Response 1: The abstract was shortened and edited, taking into account the Reviewer comments.
Point 2: The term 'non-pathogenic virus(es)' is used throughout the paper. Although this term was used also in previous papers, I found it very unfortunate. If fact, from a point of view of a biologist, there are no non-pathogenic viruses. Each virus can propagate only in the cells of its host, thus, each virus must be pathogenic, othervise it could not exist. Obviously, symptoms of infections by various viruses exprerssed by host organisms can vary significantly, from severe to mild, or even asymptomatic infections can occur. However, in no case a virus can be considered non-pathogenic. In my opinion, if the authors want to keep such a nomenclature in this paper, they should either provide such a comment in the text (if they agree with the arguments presented in this review) or provide arguments (in response to review) why they do not agree with my opinion.
Response 2: The authors of this manuscript do not question the validity of this Reviewer comment or the accepted method of defining virus pathogenicity from a biologist's perspective. We also do not wish to enter into a dispute (discussion) regarding the validity of the proposed definition.
[point #1] However, we want to emphasize and explain that with respect to lagoviruses detected and identified in rabbits (RHDVs, RCV, RCV-A1, RCV-E) and hares (EBHSV, HACV) since the mid-1980s, the term "nonpathogenic virus(es)" (considering the multitude of forms currently recognized), meaning a virus of low virulence that does not cause symptoms of disease, has long been used in the scientific literature on the occurrence of RHD worldwide (Rodak et al. 1990 J Gen Virol, 71, 1075-1080; Capucci et al. 1996 J Virol 70, 8614-8623). Further findings regarding the origin of RHDV/EBHSV lagoviruses and the mechanisms of their variability through recombination and evolution have confirmed the diversity of lagoviruses, often described by the terms "pathogenic virus" and "nonpathogenic virus." The ongoing changes within rabbit and hare caliciviruses are reflected in the multiplicity of antigenic and genetic forms identified primarily within pathogenic RHDV, but also among lagoviruses not provoking the clinical symptoms, resulting in death. This significant diversity of lagoviruses is reflected in the new taxonomy from 2017 (Le Pendu et al. 2017 J Gen Virol 98:1658-1666), which includes viruses defined as pathogenic (RHDV, RHDVa, RHDV2, EBHSV) and non-pathogenic (RCV, RCV-E1, RCV-A1), co-authored by one of the authors of our manuscript.
[point #2] Moreover, recently in some publications [(1) Cavadini et al. Two decades of occurrence of non-pathogenic rabbit lagoviruses in Italy and their genomic characterization. Sci Rep. 2024;14(1):29234; (2) Mahar et al. Benign Rabbit Caliciviruses Exhibit Evolutionary Dynamics Similar to Those of Their Virulent Relatives. J Virol. 2016, 29;90(20):9317-29; (3) Nicholson et al. Benign Rabbit Calicivirus in New Zealand. AEM, 2017, 83(11), e00090-17], the Authors use the term enterotropic benign viruses, benign rabbit caliciviruses, benign lagoviruses in parallel with the term non-pathogenic lagoviruses (probably to maintain the terminology adopted over the years to describe viruses that do not cause any clinical symptoms), departing from the unfortunate term for biologists, “non-pathogenic viruses”
[point #3] On the one hand, a comment calling for the abandonment of the use of the term "nonpathogenic" in this work would severely restrict the flow of expression, given the known data in this field and the practice of using these terms in the literature. At the same time, we wish to emphasize that in this work, we do not characterize our own virus strains with such characteristics, but merely cite known and already published information on this topic.
[point #4] On the other hand, we agree with the Reviewer comment on this issue. However, we would like to emphasize that our manuscript is not the place for a detailed discussion of the terminology used for lagoviruses, particularly those traditionally described as "nonpathogenic." In the text, we use well-established terminology that has been used in the literature for many years, which stems from the broader and more complex issue described in point 1.
Therefore, since we do not want our article to become a source of terminological polemic among virologists dealing with “benign lagoviruses,” we kindly ask you to understand our decision and accept the manuscript in its current form.
Point 3: Figure 2 is too small; especially letters are so small that it is impossible to read any name. This must be corrected; otherwise the figure is useless.
Response 3: We greatly appreciate your attention to this matter. When the manuscript was initially submitted, Figure 2 was formatted and fully legible, consistent with Figure 3. However, during the editorial processing of the manuscript, the figure was reformatted in a way that reduced its clarity, likely due to space adjustments in the layout. As a result, the reviewers received the version with the less legible figure. In the current version of the manuscript, we are providing Figure 2 in a clear and legible format.
Point 4: Table 2 is constructed against the common rules. The first column should be titled "Characteristic", while "Lagovirus europaeus strain" should be a common heading for columns 2-5.
Response 4: Table 2 has been restructured according to Reviewer suggestion.
Point 5: Chapter Conclusions should contain only real conclusions, not a summary of the obtained results. Please, restrict this chapter to a few points indicating what did you conclude based on the obtained results, not providing another description of the whole work (such a description is already provided in Materials and Methods, Results, and Discussion sections, while a summary should be presented in Abstract, rather than in Conclusions).
Response 5: The chapter “Conclusion” has been redrafted, and the conclusions improved.
We kindly request that the Reviewer accept our explanation and the changes introduced to the manuscript, and consent to publication in “Viruses”.
Best regards,
Beata Hukowska-Szematowicz, Andrzej Fitzner

Reviewer 2 Report
Comments and Suggestions for Authors
General Assessment
The manuscript addresses an important epidemiological finding: the co-circulation of RHDVa and RHDV2 strains in Poland. The dataset is relevant and the molecular/phylogenetic analyses are of interest for the field. However, the manuscript requires substantial revision in terms of English language, clarity, and structure. There are also minor scientific inconsistencies and areas where interpretation should be more cautious.
Major Comments
- Language and Grammar
 - The manuscript contains numerous grammatical errors, awkward phrasing, and literal translations from Polish. For example:
 - “The beginnings of RHD in Poland date back…” → should be “RHD was first reported in Poland in 1988…”
 - “The genome of RHDV of approximtely 7.5 kb” → “approximately”.
 - “Than carried out ELISA kit…” → incorrect; should be “An ELISA was carried out…”.

A full native English editing is needed.
- Abstract

Too long and detailed. Should be shortened and focused on objectives, methods, key results, and conclusions. Redundant expressions (e.g., “we confirmed the occurrence of the Lagovirus europaeus GI.1…” appears twice).
- Introducton

Overly lengthy, with excessive background detail. Recommendation: streamline to ~4–5 paragraphs focusing on:
- Global significance of RHD.
 - Emergence of RHDV2 and recombinants.
 - Situation in Poland.
 - Knowledge gap → aim of study.
 - Methods

Phylogenetic analysis: bootstrap threshold 70% is correct, but statement “reliable when the bootstrap value was 70%” should be softened (conventionally >70% is considered moderate support, >90% strong).
- Results

Overlap with methods (e.g., sequence length description). Consider shortening. Some results are speculative in wording (e.g., “indicates different sources of virus introduction”). This should be rephrased as “suggests” or “is consistent with”.
Discussion - Very long, repetitive, and partially reads like an introduction. Key findings should be emphasized: Detection of co-circulating GI.1a and GI.2 in same geographic area. Phylogenetic links to older Polish and international strains. Implications for epidemiology and control. Avoid long historical digressions (e.g., early 1990s Poland, extensive details on RHDV release in Australia).
Conclusion
Reasonably structured, but again too wordy. Should be shortened to a concise summary of findings and implications
Figures and Tables
Figures are informative but too dense; collapsed clades could be summarized better. Typographical inconsistencies: “starains”, “rekombinants” → must be corrected.
Minor Comments
- “approximtely” → “approximately”
 - “expession systems” → “expression systems”
 - “Than carried out ELISA kit” → “An ELISA test was performed using…”
 - “indictaed” → “indicated”
 - “Phylogenetic relationship … was considered reliable when the bootstrap value was 70%.” → rephrase as “Bootstrap values ≥70% were considered indicative of moderate support.”
 - “Rabbit Haemorrhagic Dsiease” → “Rabbit Haemorrhagic Disease”
 - “boostrap” → “bootstrap”
 - “in tern recombinants” → “in turn, recombinants”
 - “starains” → “strains”
 - “rekombinants” → “recombinants”

Author Response
Response to Reviewer 2 Comments
We sincerely thank the Reviewer for helpful comments, which greatly improved the manuscript. We have introduced changes following the Reviewer suggestions. We hope that with these changes, the manuscript will be acceptable for publication in “Viruses”.
Below, we provide a step-by-step response to the comments in the review. Changes in the manuscript are marked in yellow.
We appreciate the Reviewer's time and effort on the manuscript.
Major Comments
Point 1: Language and Grammar. The manuscript contains numerous grammatical errors, awkward phrasing, and literal translations from Polish. For example:
“The beginnings of RHD in Poland date back…” → should be “RHD was first reported in Poland in 1988…”
“The genome of RHDV of approximtely 7.5 kb” → “approximately”.
“Than carried out ELISA kit…” → incorrect; should be “An ELISA was carried out…”.
A complete native English editing is needed.
Response 1: The manuscript was proofread for linguistic accuracy by a native speaker. Linguistic errors were corrected, and corrections to identified phrases were made in accordance with the suggestions provided.
Point 2: Abstract. Too long and detailed. The document should be shortened and focused on objectives, methods, key results, and conclusions. Redundant expressions (e.g., “we confirmed the occurrence of the Lagovirus europaeus GI.1…” appears twice).
Response 2: The abstract was shortened and edited, taking into account the Reviewer comments.
Point 3: Introduction. The text is overly lengthy, with excessive background detail. Recommendation: streamline to ~4–5 paragraphs focusing on: Global significance of RHD; Emergence of RHDV2 and recombinants; Situation in Poland; Knowledge gap → aim of study.
Response 3: The "Introduction" chapter has been shortened, and information about RHDV2 in Poland has been added. However, we provide information on recombination in the discussion, and in view of the Reviewer comments that the introduction is too long, we do not repeat it in the introduction.
Point 4: Methods. Phylogenetic analysis: bootstrap threshold 70% is correct, but statement “reliable when the bootstrap value was 70%” should be softened (conventionally >70% is considered moderate support, >90% strong).
Response 4: Text in a sentence (chapter Material and Methods): “The phylogenetic relationship between the analyzed sequences was considered reliable when the bootstrap value was 70%. Changed to “Bootstrap values ≥70% were considered indicative of moderate support”
Point 5: Results. Overlap with methods (e.g., sequence length description). Consider shortening. Some results are speculative in wording (e.g., “indicates different sources of virus introduction”). This should be rephrased as “suggests” or “is consistent with”.
Response 5: We have partially taken the Reviewer suggestions into account in the “Results” section. However, we believe that the information mentioned by the Authors is very important in this part of the manuscript, and therefore we would prefer not to remove it.
Point 6: Discussion. Very long, repetitive, and partially reads like an introduction. Key findings should be emphasized: Detection of co-circulating GI.1a and GI.2 in same geographic area. Phylogenetic links to older Polish and international strains. Implications for epidemiology and control. Avoid long historical digressions (e.g., early 1990s Poland, extensive details on RHDV release in Australia).
Response 6: We have shortened the Discussion; however, we would prefer not to reduce it substantially. In our opinion, the broader context, including references to the epidemiology of RHDV in Poland as well as selected international examples, is essential for a proper understanding of the significance of our findings. These elements not only place our study within the framework of the long-term evolution of Lagovirus europaeus but also highlight the epidemiological importance of the simultaneous circulation of different genotypes. Therefore, although we have reduced specific repetitive passages, we have retained the historical perspective, as it supports the interpretation of our results and their relevance for disease monitoring, control, and virus evolution.
Point 7: Conclusion. Reasonably structured, but again too wordy. Should be shortened to a concise summary of findings and implications
Response 7: The chapter “Conclusion” has been redrafted, and the conclusions improved.
Point 8: Figures and Tables. Figures are informative but too dense; collapsed clades could be summarized better. Typographical inconsistencies: “starains”, “rekombinants” → must be corrected.
Response 8: At the Reviewer suggestion, typographical inconsistencies have been corrected. We also want to clarify that when the manuscript was initially submitted, Figure 2 was formatted and fully legible, consistent with Figure 3. However, during the editorial processing of the manuscript, the figure was reformatted in a way that reduced its clarity, likely due to space adjustments in the layout. As a result, the reviewers received the version with the less legible figure. In the current version of the manuscript, we are providing Figure 2 in a clear and readable format.
Point 9: Minor Comments.
“approximtely” → “approximately”
“expession systems” → “expression systems”
“Than carried out ELISA kit” → “An ELISA test was performed using…”
“indictaed” → “indicated”
“Phylogenetic relationship … was considered reliable when the bootstrap value was 70%.” → rephrase as “Bootstrap values ≥70% were considered indicative of moderate support.”
“Rabbit Haemorrhagic Dsiease” → “Rabbit Haemorrhagic Disease”
“boostrap” → “bootstrap”
“in tern recombinants” → “in turn, recombinants”
“starains” → “strains”
“rekombinants” → “recombinants”
Response 9: At the Reviewer suggestion, all the minor comments mentioned above have been corrected.
We kindly request that the Reviewer accept our explanation and the changes introduced to the manuscript, and consent to publication in “Viruses”.
Best regards,
Beata Hukowska-Szematowicz, Andrzej Fitzner

Round 2
Reviewer 1 Report
Comments and Suggestions for Authors
The authors responded to all my comments, and I do not have any other concerns. Just for the future, it would be great if the nomenclature regarding "non-pathogenic vs. benign viruses" (I stronly prefer the latter name) was unified among virologists which are representatives of biological sciences, veterinary medicine and other fields related to viruses. Of course, this not the aim of this paper which are recommend for publication as it is. Perhaps a commentary/review article in this matter would be desirable.